# Design of a Biaxial High-G Piezoresistive Accelerometer with a Tension–Compression Structure

**DOI:** 10.3390/mi14081492

**Published:** 2023-07-25

**Authors:** Peng Wang, Yujun Yang, Manlong Chen, Changming Zhang, Nan Wang, Fan Yang, Chunlei Peng, Jike Han, Yuqiang Dai

**Affiliations:** 1School of Mechanical Engineering, Shaanxi University of Technology, Hanzhong 723001, China; wangpeng1851@163.com (P.W.);; 2Shaanxi Key Laboratory of Industrial Automation, Hanzhong 723001, China; 3Engineering Research Center of Manufacturing and Testing for Landing Gear and Aircraft Structural Parts, Universities of Shaanxi Province, Hanzhong 723001, China

**Keywords:** tension–compression structure, biaxial, high-g, accelerometer

## Abstract

To meet the measurement needs of multidimensional high-g acceleration in fields such as weapon penetration, aerospace, and explosive shock, a biaxial piezoresistive accelerometer incorporating tension–compression is meticulously designed. This study begins by thoroughly examining the tension–compression measurement mechanism and designing the sensor’s sensitive structure. A signal test circuit is developed to effectively mitigate cross-interference, taking into account the stress variation characteristics of the cantilever beam. Subsequently, the signal test circuit of anti-cross-interference is designed according to the stress variation characteristics of the cantilever beam. Next, the finite element method is applied to analyze the structure and obtain the performance indices of the range, vibration modes, and sensitivity of the sensor. Finally, the process flow and packaging scheme of the chip are analyzed. The results show that the sensor has a full range of 200,000 g, a sensitivity of 1.39 µV/g in the X direction and 1.42 µV/g in the Y direction, and natural frequencies of 509.8 kHz and 510.2 kHz in the X and Y directions, respectively.

## 1. Introduction

Since the 21st century, multiaxis accelerometers have been critical components in modern control systems, such as industrial equipment operation, attitude recognition, and weapon-aiming adjustment systems, and have become one of the major research focuses. Currently, multiaxis accelerometers include combined, integrated, and single-structure multiaxis accelerometers [1,2].

A single-structure multiaxis sensor is capable of measuring acceleration in different directions using a single structure and has the advantages of small size, ease of processing, and high accuracy. Therefore, it has gained widespread attention in the field of high-g acceleration measurement. On the one hand, the research on single-structure multiaxis accelerometers aims to meet application requirements through the design of different sensitive structures [3,4]. For example, Hitachi Ltd. in Japan has designed a ring-beam three-axis high-shock acceleration sensor that can withstand 20,000 g high shock, with sensitivities of 0.42, 0.40, and 0.33 mV/g for the X, Y, and Z axes, respectively [5]. Yonsei University in Korea has proposed an eight-beam three-axis piezoresistive high-g accelerometer, with sensitivities of 0.2433, 0.1308, and 0.3068 mV/g for the X, Y, and Z axes, respectively, under 20,000 g impact [6]. Xi’an Jiaotong University has developed a hinge-connected dual-axis accelerometer with a self-supported piezoresistive beam that has the characteristics of force amplification and axial deformation, and an average longitudinal stress of 32.473 MPa, with a measurement sensitivity of 0.198 mV/g [7]. On the other hand, reducing the cross-axis interference of the sensor measurement and improving the measurement accuracy are also major challenges. There are mainly two solutions to this problem. One is to reduce cross-axis interference through a structural optimization design [8,9]. For example, the University of Liverpool in the UK has gradually optimized a disk-shaped accelerometer to obtain a cross-sensitive structure, which reduces the cross-axis sensitivity to 18.1% compared to the circular structure, while the sensitivity on the Z-axis is increased by 76% [10]. China Jiliang University has studied the structure of a dual-axis micromechanical accelerometer, with a cross-sensitive sensitivity of 1.92% in the X-axis under Y-axis acceleration and 1.55% in the Y-axis under X-axis acceleration [11]. The second solution is to use the layout of the piezoresistive resistors to theoretically reduce cross-axis interference through Wheatstone bridge circuits. For example, North University of China has used an eight-resistor layout to test a piezoresistive sensor with a cross-beam structure, and the cross-axis sensitivity of nonmeasurement axes to measurement axes is reduced to 1.1% [12]. Shaanxi University of Technology has reasonably arranged the piezoresistive resistors of an eight-beam sensor, and the results show that the X-axis cross-axis interference is 0.86% and the Y-axis lateral cross-axis interference is 0.77%, effectively reducing cross-axis interference [13].

Based on the above research and the requirements of high-g accelerometers in applications such as weapon penetration, aerospace, and explosive shock, a single-structure sensor design method is proposed in this paper. A tension–compression measurement mechanism is used to design a sensor structure that can meet the application requirements of high-g dual-axis measurement. Additionally, the use of a resistor layout reduces cross-interference and improves measurement accuracy.

## 2. Sensor Analysis and Design

### 2.1. Tension–Compression Measurement Mechanism

Figure 1 shows a diagram of the deformation and stress changes in a tension–compression structure. The cantilever beam has a length, width, and thickness of *l*, *b*, and *h*, respectively, and the weight of the mass block is m. Assuming that the mass block is rigid and does not deform, the specific analysis is as follows:

According to the mechanics of materials, the cantilever beam on the left undergoes tensile deformation, and the stress magnitude is given as follows:(1)$$\sigma_{L}\left(x\right)=\frac{ma}{2bh}$$

The cantilever beam on the right undergoes compressive deformation, and the stress magnitude is given as follows:(2)$$\sigma_{R}\left(x\right)=-\frac{ma}{2bh}$$

Based on the analysis, it is known that in a tension–compression structure, the cantilever beam experiences tensile stress on one side and compressive stress on the other. These stresses are equal in magnitude but opposite in direction. Consequently, a resistor can be placed on top of the cantilever beam without being affected by its position, as the stress distribution remains consistent throughout the beam. This characteristic enables the production of high-quality resistors with low process requirements at any location along the cantilever beam.

### 2.2. Sensor Design

Based on the tension–compression measurement mechanism, a high-g biaxial accelerometer is designed, as illustrated in Figure 2a. The sensor structure exhibits central symmetry and comprises a frame, eight cantilever beams, a step structure, and a mass block. The frame functions as a fixed support, and the center of mass of the cantilever beam is located on the same plane as the center of mass of the mass block. The step structure is incorporated to connect the cantilever beam with the central mass block, preventing structural flipping. The design of the step structure takes into account the actual processing difficulty, as depicted in Figure 2b. A locally magnified view of the step structure shows that the cantilever beam is suspended in the middle of the structure. If the step structure is not processed, the cantilever beam needs to be directly etched on the side of the mass block, which is a more difficult task. At the same time, by connecting with the step structure, the contact area with the mass block is increased, which increases the stiffness of the structure and improves the natural frequency of the sensor. The size of the sensor is as follows: proof block 1420 μm × 1420 μm × 400 μm (length × width × height), cantilever beam 180 μm × 200 μm × 40 μm (length × width × height), step structure width of 100 μm, with a frame width of 500 μm.

A piezoresistor is made on the cantilever beam, and the piezoresistor is selected to be distributed on the (100) crystal surface along the [110] crystal direction. By considering the stress distribution law of the tension–compression structure, the position of the resistor was determined, as shown in Figure 3a. The resistors (*R*_1_, *R*_2,_ *R*_3_, and *R*_4_) designed on the four cantilever beams parallel to the X-axis are interconnected to form a Wheatstone bridge for measuring the acceleration along the X-axis. The resistors (*R*_5_, *R*_6_, *R*_7_, and *R*_8_) designed on the four cantilever beams parallel to the Y-axis are interconnected to form another Wheatstone bridge for measuring the acceleration along the Y-axis, achieving the measurement of acceleration in the X and Y directions of the plane. The size of a single resistor is shown in Figure 3b, and it consists of a resistor strip P−, ohmic contact P+, and a contact hole.

### 2.3. Test Circuit

This article employs a semi-closed-loop Wheatstone bridge circuit as the signal conversion circuit. Each resistor on the bridge arm is grounded, and two independent output terminals, *U*_1_ and *U*_2_, are utilized to prevent cross-interference. The connection of the Wheatstone bridge circuit is illustrated in Figure 4.

*R*_1_, *R*_2_, *R*_3_, and *R*_4_ constitute a Wheatstone bridge, with *U*_1_ as its output voltage. Similarly, *R*_5_, *R*_6_, *R*_7_, and *R*_8_ form another Wheatstone bridge, with *U*_2_ as its output voltage [14].
(3)$$U_{1}=E(\frac{R_{1}}{R_{1}+R_{3}}-\frac{R_{4}}{R_{2}+R_{4}})$$
(4)$$U_{2}=E(\frac{R_{5}}{R_{5}+R_{7}}-\frac{R_{8}}{R_{8}+R_{6}})$$

When the Wheatstone bridge circuit is in a balanced state, meaning the output voltage is zero, it signifies that the resistances of the four arms adhere to a specific relationship expressed by a formula.
(5)$$R_{1}R_{2}=R_{3}R_{4},\;R_{5}R_{6}=R_{7}R_{8}$$

We extracted the path located in the middle of the cantilever beam’s surface and performed a simulation analysis using ANSYS. When an X-directional force is applied, the resistances of *R*_1_ to *R*_8_ change, as shown in Figure 5. Let the input voltage be *E*, and the changes in *R*_1_ to *R*_4_ and *R*_5_ to *R*_8_ be $\Delta R_{X1}$ and $\Delta R_{X2}$, respectively. Then, the output voltages *U*_1_ and *U*_2_ can be calculated as follows:(6)$$U_{1}=\left[\frac{R_{1}+\Delta R_{X1}}{\left(R_{1}+\Delta R_{X1})+(R_{3}-\Delta R_{X1}\right)}-\frac{R_{4}-\Delta R_{X1}}{\left(R_{4}-\Delta R_{X1})+(R_{2}+\Delta R_{X1}\right)}\right]E=\frac{\Delta R}{R}E$$
(7)$$U_{2}=\left[\frac{R_{5}-\Delta R_{X2}}{\left(R_{5}-\Delta R_{X2})+(R_{7}-\Delta R_{X2}\right)}-\frac{R_{8}-\Delta R_{X2}}{\left(R_{8}+\Delta R_{X2})+(R_{6}-\Delta R_{X2}\right)}\right]E=0$$

When a Y-directional force is applied, the resistances of *R*_1_ to *R*_8_ change, as shown in Figure 6. Let the input voltage be *E*, and the changes in *R*_1_ to *R*_4_ and *R*_5_ to *R*_8_ be $\Delta R_{Y1}$ and $\Delta R_{Y2}$, respectively. Then the output voltages *U*_1_ and *U*_2_ can be calculated as follows:(8)$$U_{1}=\left[\frac{R_{1}-\Delta R_{Y1}}{\left(R_{1}-\Delta R_{Y1})+(R_{3}-\Delta R_{Y1}\right)}-\frac{R_{4}-\Delta R_{Y1}}{\left(R_{2}-\Delta R_{Y1})+(R_{4}-\Delta R_{Y1}\right)}\right]E=0$$
(9)$$U_{2}=\left[\frac{R_{5}+\Delta R_{Y2}}{\left(R_{5}+\Delta R_{Y2})+(R_{7}-\Delta R_{Y2}\right)}-\frac{R_{8}-\Delta R_{Y2}}{\left(R_{6}+\Delta R_{Y2})+(R_{8}-\Delta R_{Y2}\right)}\right]E=\frac{\Delta R}{R}E$$

When a Z-directional force is applied, the resistances of *R*_1_ to *R*_8_ change, as shown in Figure 7. Let the input voltage be *E*, and the changes in *R*_1_ to *R*_8_ be $\Delta R_{Z}$, respectively. Then the output voltages *U*_1_ and *U*_2_ can be calculated as follows:(10)$$U_{1}=\left[\frac{R_{1}-\Delta R_{Z}}{\left(R_{1}-\Delta R_{Z})+(R_{3}-\Delta R_{Z}\right)}-\frac{R_{4}-\Delta R_{Z}}{\left(R_{2}-\Delta R_{Z})+(R_{4}-\Delta R_{Z}\right)}\right]E=0$$
(11)$$U_{2}=\left[\frac{R_{5}+\Delta R_{Z}}{\left(R_{5}+\Delta R_{Z})+(R_{7}-\Delta R_{Z}\right)}-\frac{R_{8}-\Delta R_{Z}}{\left(R_{6}+\Delta R_{Z})+(R_{8}-\Delta R_{Z}\right)}\right]E=0$$

In summary, the designed measurement circuit in this article can theoretically measure the acceleration in the X and Y directions separately without being affected by acceleration in other directions. This means that *U*_1_ only outputs a voltage signal proportional to the acceleration in the X direction, while *U*_2_ only outputs a voltage signal proportional to the acceleration in the Y direction. Therefore, the test circuit is designed reasonably. Additionally, when the structure is subjected to acceleration in the Z direction, the eight resistors on the Wheatstone bridge are in the same stress area, which results in the same and nearly zero stress changes. As a result, *U*_1_ and *U*_2_ have no output in the Z direction. This verifies the feasibility of the resistor layout method proposed. With the designed test circuit, cross-interference from nonmeasurement directions can be eliminated, ensuring the accuracy of the measurement.

## 3. Simulation Analysis

### 3.1. Modal Analysis

Modal analysis is a method used to determine the vibration characteristics of a structure by solving its natural frequencies and vibration modes. The goal is to avoid resonance or vibration at specific frequencies. Modal analysis can be divided into free and constrained modal analysis. This article uses constrained modal analysis, with fixed supports added to the silicon frame and the program-controlled method to extract the first six natural frequencies and vibration modes of the sensor structure, as shown in Figure 8.

After the analysis, it is observed that the first six vibration modes of the structure correspond to its six degrees of freedom. The first three modes are classified as low-frequency modes, while the remaining three modes are categorized as high-frequency modes: specifically, the vibration along the Z-axis (Mode 1) and rotational vibration around the Z-axis (Mode 6), with frequencies of 112.83 and 677.9 kHz, respectively; the vibration along the X-axis (Mode 4) and rotational vibration around the X-axis (Mode 2), with frequencies of 509.80 and 177.35 kHz, respectively; and the vibration along the Y-axis (Mode 5) and rotational vibration around the Y-axis (Mode 3), with frequencies of 510.23 and 180.03 kHz, respectively. Notably, the frequency differences between the various directions of vibration and rotation are relatively large, while the mutual interference is minimal. This indicates that the structure is designed reasonably.

### 3.2. Harmonic Response Analysis 

The harmonic response is the steady-state response of a linear system to harmonic excitation, which can reflect the motion characteristics of the structure under different frequency harmonic loads. The working environment of the sensor in this paper contains many high-frequency interference signals. When these frequencies are consistent with the natural frequency of the sensor, resonance can occur, affecting the service life of the sensor. Therefore, harmonic response analysis is required for the sensor. 

According to the modal analysis of the sensor, the frequency range of the harmonic response is set to 0–700 kHz, with 100 sample points and a frequency interval of 7 kHz. A peak sine acceleration load of 200,000 g is applied, and a fixed constraint is added to the frame. Harmonic response analysis is carried out on the structure in the X, Y, and Z directions. X, Y, and Z reach the maximum amplitudes around 510 kHz, 510 kHz, and 110 kHz, respectively, that is, the resonance phenomenon occurs, and the frequency of these three points is close to the fourth-, fifth-, and first-order natural frequencies, consistent with the motion of the modal vibration.

### 3.3. Sensitivity Analysis 

Sensor sensitivity refers to the relationship between the output signal and the change in the measured physical quantity. For a piezoresistive accelerometer, its sensitivity *S* is calculated by the following formula:(12)$$S=\frac{U}{a}=\frac{\Delta R}{R}E*\frac{1}{a}$$

Whereas it is known that the piezoresistive effect is given by the following:(13)$$\frac{\Delta R}{R}=\frac{1}{2}\pi_{44}(\sigma_{l}-\sigma_{t})$$
where $\pi_{44}$ is the piezoresistive coefficient of silicon, which is 138.1 × 10^−11^ m^2^/N, $\sigma_{l}$ is the longitudinal stress, and $\sigma_{t}$ is the transverse stress.

Based on the above analysis, the sensitivity of a piezoresistive accelerometer can be measured by detecting the stress changes on a cantilever beam. Therefore, in this paper, three paths along the direction of the piezoresistive element are selected in the region of the resistor, and the paths *R*_1_, *R*_2_, *R*_3_, and *R*_4_ are shown in Figure 9a, and the paths *R*_5_, *R*_6_, *R*_7_, and *R*_8_ are shown in Figure 9b. 

Under the acceleration shock in the X direction, stress differences are obtained on paths *R*_1_ to *R*_4_. The simulation results are substituted into Equation (12), assuming an input voltage of 5 V, to calculate the output voltage, as shown in Table 1.

Linear regression is performed on the voltage output data, and a normal distribution is used for correlation analysis. The regression equation is as follows: (14)U=0.00139 mV/g×a+0.045 mV

Under the acceleration shock in the Y direction, stress differences are obtained on paths *R*_5_ to *R*_8_. The simulation results are substituted into Equation (12), assuming an input voltage of 5 V, to calculate the output voltage, as shown in Table 2.

Linear regression is performed on the voltage output data, and a normal distribution is used for correlation analysis. The regression equation is as follows:(15)U=0.00142 mV/g×a+0.10 mV

Based on the results, it can be concluded that within the acceleration range of 50,000 g to 200,000 g, the sensitivity in the X direction is 1.39 µV/g and the sensitivity in the Y direction is 1.42 µV/g, which meets the requirements for the design of high-g sensors.

## 4. Process Flow and Packaging Scheme

### 4.1. Process Flow

This paper utilizes silicon micromachining technology to make the sensor chip, and the process is shown in Figure 10:(1)SOI wafers: SOI material is used, with single-crystal silicon material on the top and bottom layers and a silicon dioxide layer in the middle. SOI material has the advantages of high-temperature resistance and low-power consumption, which can meet the special requirements for aerospace, aviation, and explosion impact.(2)Deep reactive ion etching (DRIE): DRIE is performed on both sides of the wafer to form a distributed mass block structure.(3)Surface growth of silicon dioxide: Thermal oxidation is used to generate a silicon dioxide layer as an insulating layer for the subsequent processing of the resistor.(4)Production of P− and P+: In the first lithography step, boron ions are injected to form a P-type heavily doped region, which is the ohmic contact. In the second lithography step, boron ions are injected to form a P-type lightly doped region, which is the resistor.(5)Surface growth of silicon nitride: A 100 nm silicon nitride layer is formed on the surface using low-pressure chemical vapor deposition technology to protect the processed resistor.(6)Production of contact holes, wire leads, and pads: In order to connect the resistor to the chip surface, a third lithography step is performed to create contact holes at the base of the resistor. The boundary of the contact hole should be 2 μm smaller than that of the resistor strip. A 1 μm thick aluminum film is sputtered on the surface using PVD, and aluminum electrodes and pads are formed by lithography, followed by 30 min of metallization at 500 °C.(7)Inductively coupled plasma (ICP): ICP etching is performed on the front of the silicon wafer to release the structure.(8)Silicon–glass bonding technology: The silicon wafer and glass are bonded on a 300 μm thick Pyrex 7740 glass, with a bonding depth of 20 μm, forming an isolation and protective layer for the sensor chip.

### 4.2. Packaging Scheme

Considering the fragility of the chip material, it is essential to provide necessary packaging protection before practical applications and even basic performance testing can be conducted. This is especially crucial in fields such as military weaponry and aerospace, where the working environment is complex and variable, and higher packaging requirements are necessary [15]. The following outlines the packaging process utilized in this article:Chip bonding

Chip bonding refers to the process of securely affixing the chip onto the PCB. In this paper, adhesive bonding is employed as the chosen method due to its simplicity, speed, and cost-effectiveness. The chip bonding process involves the precise placement of a small amount of epoxy resin adhesive onto the substrate. Subsequently, the back of the chip is aligned and attached to the PCB, following the direction indicated by the arrow. The attachment position is determined by the square drawn on the PCB beforehand. During the attachment process, care should be taken to ensure that the pads on the chip correspond to those on the PCB for easy wire bonding in the next step. Finally, the chip and the substrate are bonded together by curing the epoxy resin at a temperature of 150 °C to 250 °C. Figure 11 shows the bonding process between the PCB and the chip.

2.Lead bonding

The thermal ultrasonic gold-ball bonding method is adopted in this paper, which consists of two stages, as shown in Figure 12. 

In the initial bonding stage, a gold wire is threaded through the central hole of a capillary tool. The temperature at the end of the gold wire is elevated, causing it to melt and form a ball. The clamp holding the gold wire is then opened to manipulate the wire. Heat, pressure, and ultrasonic vibration are applied, and as the capillary tool contacts the heating pad, the melted gold ball bonds to the pad. Once the first ball bonding is accomplished, the capillary tool is raised to a position slightly higher than the premeasured loop height and moved to the second bonding pad to form a wire loop. In the second bonding stage, heat, pressure, and ultrasonic vibration are applied to the capillary tool, and the second gold ball is formed and pressed onto the PCB pad to complete the bonding. After the wire bonding, the gold wire is pulled to create a tail. The clamp of the capillary tool is then tightened, and the excess gold wire is cut, finalizing the second wire bonding process.

3.Packaging scheme

This article uses a stainless steel housing that has a high hardness, impact resistance, corrosion resistance, and high-temperature resistance. It provides a closed environment for the sensor to protect the fragile chip. Figure 13 shows the schematic diagram of the overall packaging of the dual-axis high-g acceleration sensor. The sensor chip, PCB, and housing are bonded together to form a self-supporting structure, which enables the overall structure to withstand impact and accurately reflect external vibrations.

## 5. Conclusions

This paper presents a comprehensive analysis of a biaxial high-g accelerometer with a tension–compression structure, covering various aspects such as the measurement mechanism, structural design, test circuit, performance indices, process flow, and packaging scheme. The key contributions of this work are outlined as follows:(1)The stress variation characteristics of the deformation of a cantilever beam under tension–compression are studied to determine the sensor’s sensitive structure. A test circuit resistant to cross-interference is designed to enhance measurement accuracy.(2)Modal analysis, harmonic response analysis, and sensitivity analysis of the designed high-g biaxial accelerometer are carried out using finite element analysis. The results show that the sensor has a range of 200,000 g, a minimum natural frequency of 112.83 kHz, and resonance frequencies of 509.8 kHz and 510.23 kHz in the X and Y directions, respectively. A relationship model between the output voltage and the acceleration of the sensor is established using MATLAB, with a measurement sensitivity of 1.39 µV/g in the X direction and 1.42 µV/g in the Y direction.(3)The process flow of the sensor is developed, with key process steps including photo etching, ion implantation, metal sputtering, thin film deposition, release structuring, and silicon–glass bonding. The chip is then bonded using an adhesive, and wire bonding was completed using a hot ultrasonic gold-wire-ball bonding method. Finally, the sensor chip is packaged using the combination of a metal casing and PCB. The proposed biaxial high-g accelerometer has potential applications in the aerospace, military, and transportation sectors.

## Figures and Tables

**Figure 1 micromachines-14-01492-f001:**
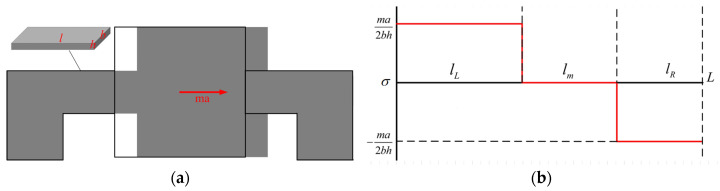
Measurement mechanism of a piezoresistive accelerometer with a tensile–compression structure: (**a**) Diagram of the deformation; (**b**) Diagram of the stress.

**Figure 2 micromachines-14-01492-f002:**
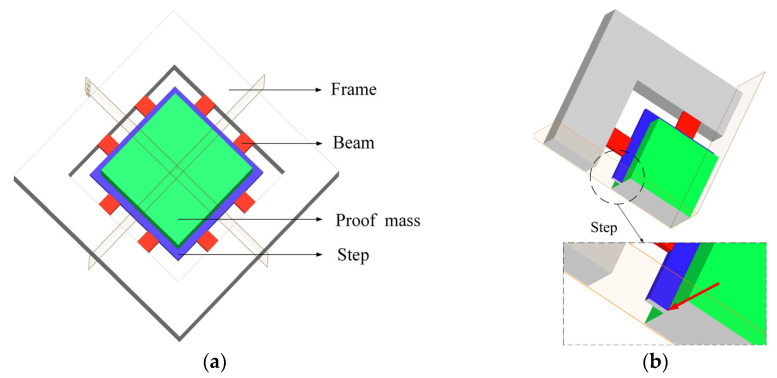
Diagram of the sensor structure: (**a**) Overall structure diagram; (**b**) Step structure.

**Figure 3 micromachines-14-01492-f003:**
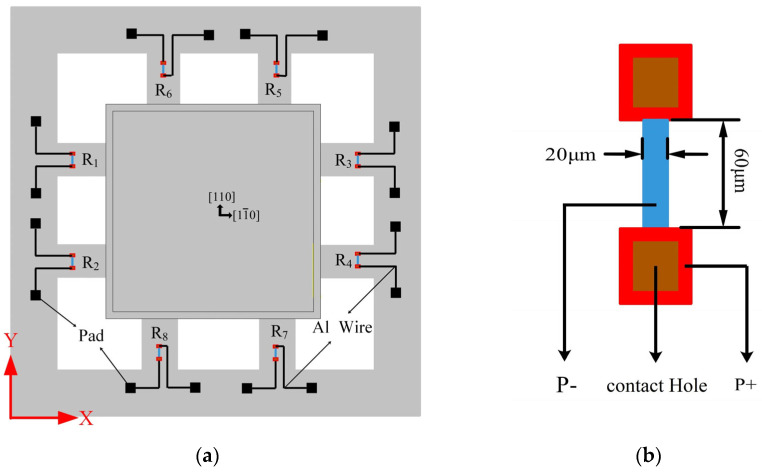
Diagram of the piezoresistor: (**a**) Piezoresistor position; (**b**) Piezoresistor size.

**Figure 4 micromachines-14-01492-f004:**
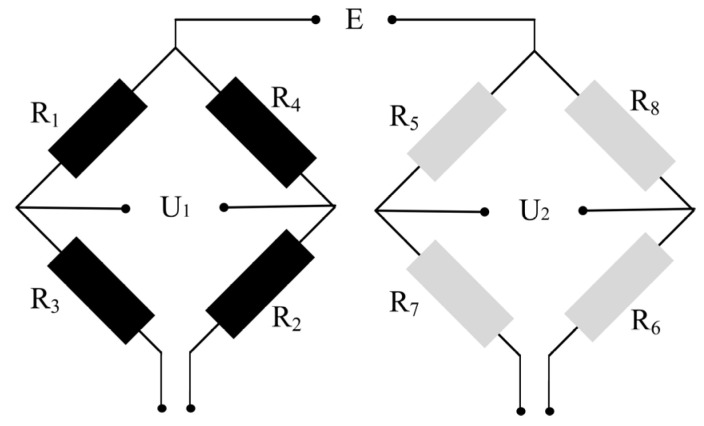
Wheatstone circuit.

**Figure 5 micromachines-14-01492-f005:**
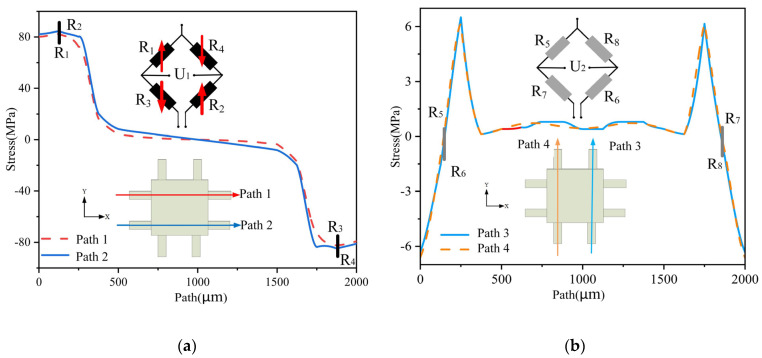
Diagram of path stress in X direction: (**a**) *R*_1_–*R*_4_ path stress; (**b**) *R*_5_–*R*_8_ path stress.

**Figure 6 micromachines-14-01492-f006:**
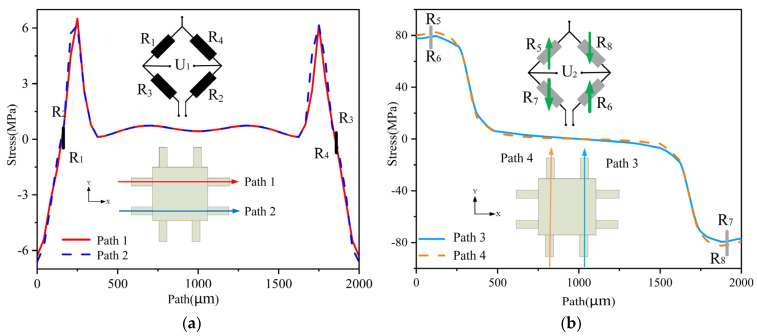
Diagram of path stress in Y direction: (**a**) *R*_1_–*R*_4_ path stress; (**b**) *R*_5_–*R*_8_ path stress.

**Figure 7 micromachines-14-01492-f007:**
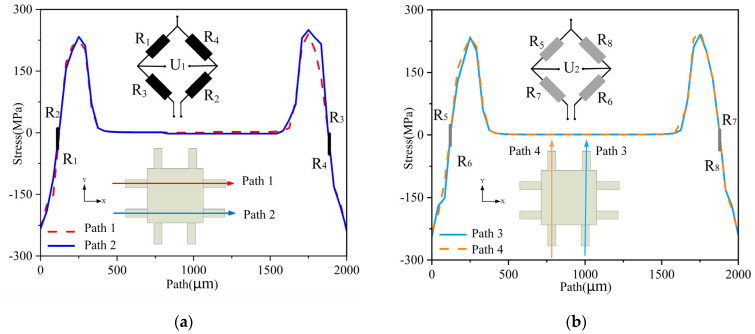
Diagram of path stress in Z direction: (**a**) *R*_1_–*R*_4_ path stress; (**b**) *R*_5_–*R*_8_ path stress.

**Figure 8 micromachines-14-01492-f008:**
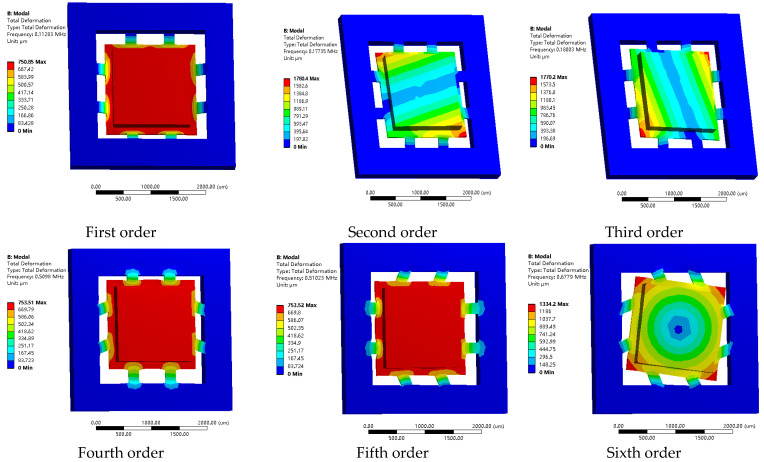
Mode of vibration diagram.

**Figure 9 micromachines-14-01492-f009:**
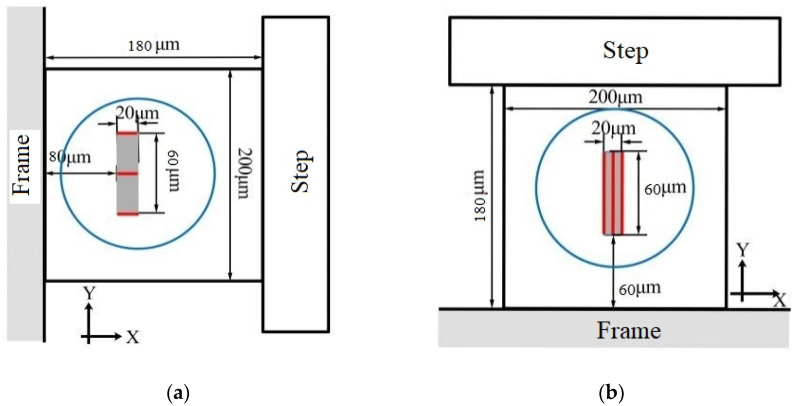
Stress path diagram: (**a**) *R*_1_ to *R*_4_; (**b**) *R*_5_ to *R*_8_.

**Figure 10 micromachines-14-01492-f010:**
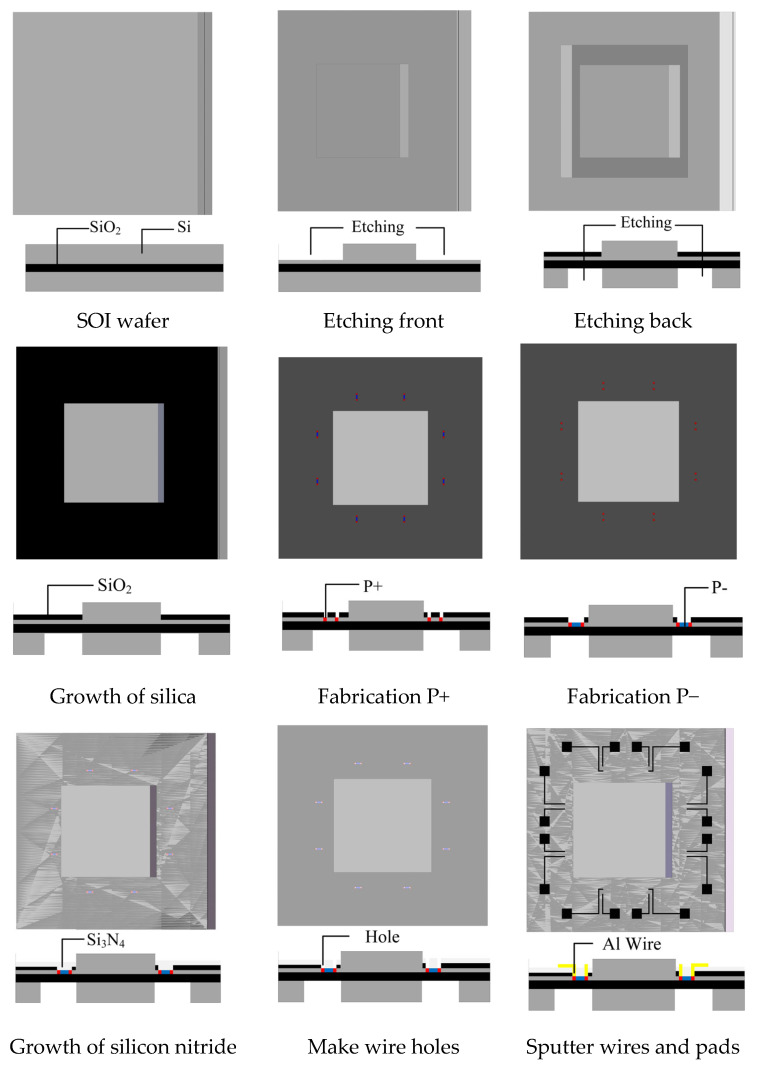
Process of the accelerometer.

**Figure 11 micromachines-14-01492-f011:**
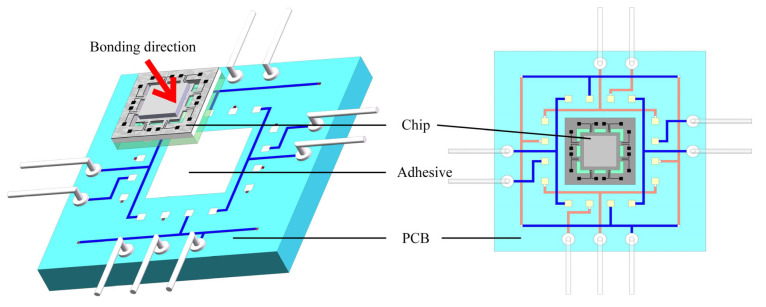
Diagram of the adhesive bonding.

**Figure 12 micromachines-14-01492-f012:**
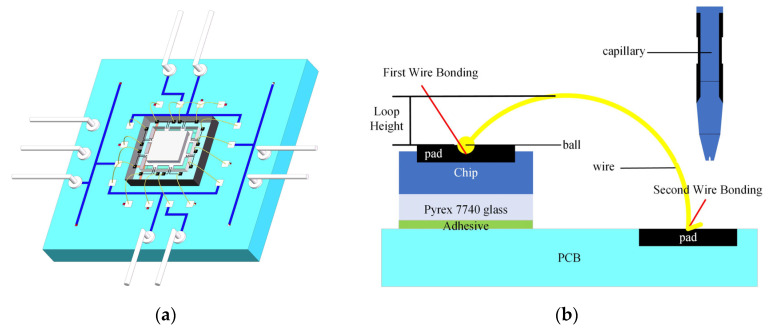
Lead bonding diagram: (**a**) 3D model diagram; (**b**) Thermal ultrasonic gold-ball bonding method.

**Figure 13 micromachines-14-01492-f013:**
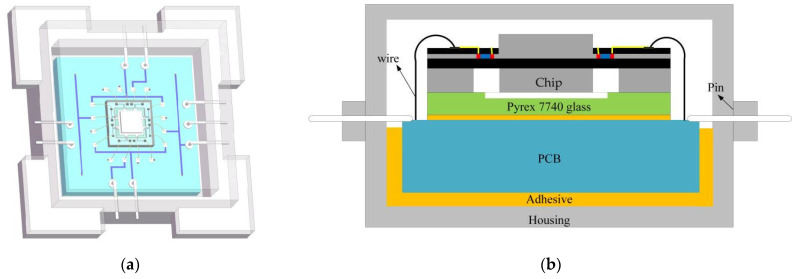
Diagram of the accelerometer package: (**a**) 3D model diagram; (**b**) Packaging scheme.

**Table 1 micromachines-14-01492-t001:** Voltage output variation in the X direction.

**A** **cceleration (** × **10^4^ m/s^2^)**	5 g	8 g	10 g	13 g	15 g	18 g	20 g
**S** **tress difference (MPa)**	20.1	32.2	40.4	52.6	60.4	72.5	80.7
**Voltage (mV)**	69.4	111.2	139.5	181.6	208.5	250.3	278.6

**Table 2 micromachines-14-01492-t002:** Voltage output variation in the Y direction.

**A** **cceleration (** × **10^4^ m/s^2^)**	5 g	8 g	10 g	13 g	15 g	18 g	20 g
**S** **tress difference (MPa)**	20.6	32.7	41.2	53.6	61.4	74.0	82.1
**Voltage (mV)**	71.1	112.9	142.2	185.1	212.0	255.5	283.5

## Data Availability

The data presented in this study are available on request from the corresponding author.

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
