# Peer review of "Design of a Biaxial High-G Piezoresistive Accelerometer with a Tension–Compression Structure"

_micromachines, 2023, doi:10.3390/mi14081492_

Round 1

Reviewer 1 Report

Line 18: Dont seperate (line break) value and unit (200000 g)

Figure 1a: You mention in the text the l, b and h of the structure shown in Figure 1a. Would help the understanding if you add these descriptions into the figure

Figure 1a: The images shows the stress path along the x axis. In the image you see a gradient (varaition of color) where as in the graph 1b you don´t see that, you just have a constant stress value. 

Figure 1b: add the description of the y axis (stress or Sigma)

(It might increase understanding if you put the scatch and the graph on top of each other)

I would like to suggest to add information about the dimensions of the Sensor (e.g. in Figure 3a)

In Figure 3b) rename Hole to "contact hole"

Add an coordinate System into figure 3 as you refer to x and y direction serveral time in your text this helps understanding. 

(Do the same for Figure 5 and 6)

What type of stress is shown in Figure 5 and 6? Is it van Mise stress?

Please mention in the text  or highlight that the figures 5a and 5b (as well as 6a and 6b) have different scaling on the y axis. 

It is not mentioned in the texte that these graphs in figure 5 and 6 are simulation results. 

(Can you provide any explanation why the graphs differ in one plot PAth1 vs Path2? Is it because of the mesh that was used in the model?)

Line 145: You are right, that all resistors are exited in the same dirction, but the transversal resitors R1-R4 will differ in the direction of change compared to R5-R8!

For a better visibility an increase of the legends in figure 8 would help - the numbers are barly readable.

If possible apply the same scaling of the Deformation! 

Please check if the frequencies mentioned in Lines 163-169 really match the images. 

In the images you use MHz and in the text kHz - if possible choose one magnitude.

Line 198: Please add the source where you have the value of the piezoresistive coeeficents from. 

In Figure 11 and Figure 12 the formulas are bearly readable. Please increase. 

You could provide some description of the manufacturing process shown in section 4.1.

It would be specially interesting how the etching was performed and some information about the depth of the etches. 

It would be great to learn more about the testing of this sensor. How can this sensor be testet? 
Assuming that the sensor does not survice the test, why is there this quite costly package and how does the readout work?

Has the package be realized yet? 

Author Response

cover letter

 (micromachines-2482298 )

Dear reviewer 1,

On behalf of my co-authors, I would like to thank you very much for giving us an opportunity to revise our manuscript entitled “Design of biaxial high g piezoresistive accelerometer with tension-compression structure”. You are highly insightful and enable us to greatly improve the quality of our manuscript.

In accordance with the comments we received, careful modifications have been made. All changes have been marked using blue highlight in the revised version. We have uploaded following files in resubmission process: (1) cover letter and detailed responses to reviewer, (2) revised manuscript with modifying trace kept, (3) The final version manuscript.

We hope the revisions in the revised manuscript and our accompanying responses will be sufficient to make our manuscript eligible for publication. Sincere thanks again to your conscientious work and precious time. We are looking forward to hearing from you at your earliest convenience.

Correspondence and email about the paper can be directed to Yujun Yang at the following address.

Yujun Yang

School of Mechanical Engineering, Shaanxi University of Technology, Han zhong, Shaanxi, 723000, China.

Reviewer 2 Report

   The reviewed article discusses the principle of operation, potential characteristics, design and production technology of a dual-axis MEMS high-g accelerometer. The article is useful and will be of interest to specialists working in this field.

There are a number of remarks.

  1. In formulas (1) and (2), the parameter A is used. The text does not explain what it is (obviously, this is the cross-sectional area of the cantilever beam). Since this parameter is not used further in the text, it is better to simply exclude it from formulas (1),(2).

  2. Line 112: Instead of "Figure 2(b)" should be "Figure 3(b)".

  3.      Line 113: It should be explained what P- and P+ are.

4.      According to Figure 3(b), the piezoresistor length is 60 microns, but Figure 10 indicates this length of 70 microns. What is the correct length?

5.      Section 2.3: Figure 4 should show the complete electrical circuit with all the connections between the elements so that the formulas for calculating U1 and U2 are clear.

6.      Formula (6) is identical to formula (8), and formula (7) is identical to formula (9), although the force is applied in different directions. Authors should check these formulas.

7.      Figures 5, 6, 7 show stress diagrams, but it does not say how these diagrams were obtained. This should be clarified.

8.      Figure 9 can be excluded, since the description in the text is sufficient.

9.      Figures 11 and 12 are also uninformative, they can be combined or even excluded. The errors of parameters a,b in linear regressions can be given in formulas (14) and (15).

10.  The authors did not analyze the effect of deviations in the values of resistors R1-R8, as well as deviations in the geometry of the cantilever beams on the parameters of the accelerometer. The inclusion of such information in the text would be very desirable.

11.  Section 4.1 is too short. Comments should be given on the technological operations shown in Figure 13.

Although there are a lot of remarks, they are not critical and do not require significant revision of the article. After minor revision, the article can be published.

Author Response

cover letter

 (micromachines-2482298 )

Dear reviewer 2,

On behalf of my co-authors, I would like to thank you very much for giving us an opportunity to revise our manuscript entitled “Design of biaxial high g piezoresistive accelerometer with tension-compression structure”. You are highly insightful and enable us to greatly improve the quality of our manuscript.

In accordance with the comments we received, careful modifications have been made. All changes have been marked using blue highlight in the revised version. We have uploaded following files in resubmission process: (1) cover letter and detailed responses to reviewer, (2) revised manuscript with modifying trace kept, (3) The final version manuscript.

We hope the revisions in the revised manuscript and our accompanying responses will be sufficient to make our manuscript eligible for publication. Sincere thanks again to your conscientious work and precious time. We are looking forward to hearing from you at your earliest convenience.

Correspondence and email about the paper can be directed to Yujun Yang at the following address.

Yujun Yang

School of Mechanical Engineering, Shaanxi University of Technology, Han zhong, Shaanxi, 723000, China.

Email: [email protected].

Reviewer 3 Report

The authors designed a two-axis high-g accelerometer based on the piezoresistive effect and the tension-compression mechanism of the beam. The author briefly described this mechanism in this article and built a detection circuit based on the Wheatstone bridge circuit that can reduce signal interference from different directions. The author analyzes the modal and harmonic response of the designed structure through the finite element method and gives some performance indicators of the accelerometer. The author also gives the fabrication process and packaging method of the accelerometer. Overall, the work is relatively systematic and of value. However, some issues still need to be resolved:

1. On line 14, what is the meaning of ‘sensor of the sensor was designed’?

2. On lines 60-67, the author mentioned that most of the domestic and foreign solutions only focus on one of the two mentioned earlier. However, can this work be considered a combination of the two methods? Otherwise, it is recommended to introduce this work in another way.

3. In Figure 1(a), it is suggested to label where it is fixed in this tension-compression beam mechanism so that it is more intuitive.

4. In Figure 3, why is the direction of the resistance length not all parallel or perpendicular to the direction of the beam? Is there any deep meaning in the arrangement direction used in the figure? If so, please explain. In addition, it is recommended to add x, y, and z coordinate axes in Figure 3 to help better explain the direction specifically later.

5. It is suggested that the author carefully check formulas 8 and 9.

6. In line 162, there is an inconsistency in capitalization

7. In line 198, the usage of π to represent the piezoresistive coefficient may cause unnecessary misunderstandings.

8. In line 203, the terms such as R1 mentioned here are not shown in Figure 10.

9. In line 222, the author did not explain why only the conclusion under the range of 50000 g~200000 g was used as the accelerometer range. It is too arbitrary to simply assume that the range of their accelerometer is highly linear in this range just because they found a high degree of linearity in this range through simulation.

10. In Figure 13, the third picture is incorrect, and SiO2 is placed on it prematurely. In addition, some of the content in this figure is not clear enough and lacks relevant serial number annotations. If the author does not intend to explain it in words, it is best to express it clearly in the figure.

11. In line 287, there is a punctuation problem.

Language has to be polished by native speakers.

Author Response

cover letter

 (micromachines-2482298 )

Dear reviewer 3,

On behalf of my co-authors, I would like to thank you very much for giving us an opportunity to revise our manuscript entitled “Design of biaxial high g piezoresistive accelerometer with tension-compression structure”. You are highly insightful and enable us to greatly improve the quality of our manuscript.

In accordance with the comments we received, careful modifications have been made. All changes have been marked using blue highlight in the revised version. We have uploaded following files in resubmission process: (1) cover letter and detailed responses to reviewer, (2) revised manuscript with modifying trace kept, (3) The final version manuscript.

We hope the revisions in the revised manuscript and our accompanying responses will be sufficient to make our manuscript eligible for publication. Sincere thanks again to your conscientious work and precious time. We are looking forward to hearing from you at your earliest convenience.

Correspondence and email about the paper can be directed to Yujun Yang at the following address.

Yujun Yang

School of Mechanical Engineering, Shaanxi University of Technology, Han zhong, Shaanxi, 723000, China.

Email: [email protected].
